# Changing Patterns of Intimate Partner Violence against Pregnant Women: A Three-Year Longitudinal Study

**DOI:** 10.3390/ijerph192114397

**Published:** 2022-11-03

**Authors:** Xiao Yan Chen, Camilla K. M. Lo, Frederick K. Ho, Wing Cheong Leung, Patrick Ip, Ko Ling Chan

**Affiliations:** 1Department of Applied Social Sciences, The Hong Kong Polytechnic University, Hung Hom, Hong Kong; 2Institute of Health and Wellbeing, University of Glasgow, 1 Lilybank Gardens, Glasgow G12 8RZ, UK; 3Department of Obstetrics & Gynaecology, Kwong Wah Hospital, Kowloon, Hong Kong; 4Department of Paediatrics and Adolescent Medicine, The University of Hong Kong, Pokfulam, Hong Kong

**Keywords:** pregnant women, intimate partner violence, longitudinal design, changing patterns, Chinese

## Abstract

Intimate partner violence (IPV) against pregnant women adversely impacts women’s and infants’ health. This study aims to provide longitudinal evidence regarding how pregnant women’s exposure to IPV changes over time. Additionally, we examine the risk and protective factors associated with these changes. In total, 340 pregnant women were recruited from an antenatal clinic in Hong Kong. IPV experiences and health conditions were assessed at pregnancy and at both 4 weeks and 3 years after childbirth. The women also reported adverse childhood experiences (ACEs), their family support, and perceived partner involvement. We found IPV prevalence among the study sample decreased from 22.9% before pregnancy to 13.5% during pregnancy, 14.7% at 4 weeks after childbirth, and 11.8% at 3 years after childbirth. We further found three types of IPV: 11.8% of women had a violent relationship (VR) persistently over time from pregnancy to 3 years after childbirth, 20.6% experienced decreased IPV (DVR), and 67.6% reported a nonviolent relationship (NVR) throughout the study period. VRs were associated with more severe mental health problems and higher ACEs. Family support and partner involvement may be protective factors for decreased IPV. Our present findings highlight the importance of identifying different IPV types over time to provide targeted intervention to the most vulnerable groups.

## 1. Introduction

Intimate partner violence (IPV) against women is a serious but preventable public health problem. It typically includes physical, psychological, and sexual violence [1]. A study including women from 161 countries and areas indicated that 27% of ever-partnered women aged 15–49 years reported physical or sexual (or both) IPV in their lifetime [2]. IPV against pregnant women is of particular concern as it affects both women’s and children’s health [3]. Physical IPV during pregnancy showed a wide range (1.6–78%), as did psychological IPV (1.8–67.4%) [4]. A recent meta-analysis found a worldwide prevalence of physical, psychological, and sexual IPV during pregnancy of 9.2%, 18.7%, and 5.5%, respectively [5]. These prevalence studies provide useful information about the occurrence of IPV at the population level, but it is unclear how and whether IPV changes across individuals over time [6]. In addition, the variations in IPV prevalence across studies may be due to different timings of IPV assessment [7].

Identifying categories of IPV across time is critical for developing effective and targeted preventions and interventions [8,9], and some research efforts have been made in this area. Jackson et al. (2015) revealed that IPV decreased from 13.1% before pregnancy to 11.3% during pregnancy [10]. Another study found that about 21% of women experienced some forms of IPV during pregnancy, and the figure increased to nearly 25% in the first 4 months postpartum [11], providing additional evidence to support the idea that women’s exposure to IPV may change during the course of pregnancy and after childbirth. However, it is not likely that all women experience the same type of change in IPV over time. For instance, Johnson’s (2008) typology of IPV includes several different IPV-related relationships: women in abusive relationships are controlled and abused over a long period, whereas other women may experience relatively short-term stress-related situational IPV [12]. Regarding pregnancy-related IPV, a cross-sectional study of 426 new mothers retrospectively reported their experience of IPV before, during, and 6 months after pregnancy, and it was found that 60.8% of the mothers reported continued exposure to IPV from pregnancy into the postpartum period [13]. Furthermore, a longitudinal study (assessing IPV before pregnancy, during pregnancy, and 4 weeks after childbirth) with 1083 pregnant women identified four categories of IPV based on the participants’ experience of IPV over the study period: (a) abusive relationship (13.2%); (b) relationship with decreased violence over pregnancy (11.4%); (c) relationship with stress-related violence (9.1%); and (d) nonviolent relationship (66.3%) [14]. While these studies provide us with critical insight into changes in IPV, the follow-up periods of these studies were relatively short. A longer follow-up period is needed to improve our understanding of the trajectory of IPV.

Another critical task is to explore the correlates of these groups. Dutton’s ecological theory is a widely used model to interpret different systems related to IPV [15]. Previous reviews have indicated that exosystem (e.g., social support), microsystem (e.g., past traumatic experiences), and ontogenetic (e.g., mental health) factors are associated with IPV victimization [16,17]. Among established findings, adverse childhood experiences (ACEs) are one of the most studied factors linked to IPV. Trauma-focused theories such as the self-trauma model point out that traumatic experiences could affect an individual’s development via different mechanisms (e.g., coping strategies and distorted cognitive understandings of self and others), which may further affect intimate relationships [18]. A meta-analysis found that ACEs (e.g., witnessed violence in childhood) were risk factors for IPV among Chinese women [19]. To our knowledge, very few studies have explored whether ACEs are specifically associated with different patterns of IPV. Such work is crucial to generating a more nuanced understanding of the impact of ACEs on IPV. In addition, we will examine exosystem (family support and partner involvement) and ontogenetic (health factors) correlates across different IPV patterns. Previous work has indicated the association between IPV groups and these variables. For example, pregnant women in a long-term and violent relationship reported more severe depression, lower levels of partner involvement, and poorer social support [14]. However, few longitudinal studies have specifically examined how these factors predict different IPV categories from pregnancy to after childbirth.

This study aimed to investigate the changing patterns and associated predictors of IPV across three years among pregnant women in Hong Kong. Specifically, the first objective was to identify categories of IPV based on women’s IPV experiences from pre-pregnancy to three years after childbirth. We hypothesized that different categories of IPV would be identified. The second objective was to investigate risk and protective factors associated with the different categories. We hypothesized that women in a constantly violent relationship would have higher ACEs, poorer health conditions, lower family support, and lower partner involvement.

## 2. Materials and Methods

### 2.1. Study Design and Participants

This was a longitudinal survey conducted between 2016 and 2020 in Hong Kong. Pregnant women were recruited from the antenatal clinic at Kwong Wah Hospital, a public hospital managed by the Hospital Authority in Hong Kong. The inclusion criteria were: (1) all women attended these clinics for their first antenatal visit; (2) they were 18 years old or older; (3) they could understand written Chinese; and (4) they provided informed consent and were willing to provide contact methods for our follow-up survey.

### 2.2. Procedures

Numerous safety and security issues had to be considered. First, we provided informed consent to all participants before participating in the study and told them that they could withdraw from the survey at any time. Second, trained research staff participated in the study’s whole procedure. We first approached women in the clinical center, a safe space for participating in our survey. During the follow-ups, we asked the participants whether it was safe and appropriate to maintain privacy and confidentiality while assessing. Third, we provided the necessary information for referral and encouraged women to seek help when identified as abuse victims. The hospital was available to provide professional assistance and consultation to the identified abuse victims if needed. Fourth, data confidentiality was strictly protected. All study data are pseudonymized and stored on protected servers. Only members of the research team can access the data. Lastly, the research protocol was approved by the Institutional Review Board of the Hospital Authority Kowloon West Cluster Research Ethics Committee and the Clinical Research Ethics Review of the Hong Kong Polytechnic University (reference number: KW/FR-16-042(97-01)(3) and CRESC201905).

In 2016 (T1, about 20–24 weeks of gestation), 758 participants were recruited and completed a structured questionnaire while waiting for their first antenatal checkup. They also provided their expected delivery date and preferred method to contact them for the follow-up. In 2017 (T2, about 4 weeks after childbirth), we contacted the participants again for the follow-up survey. In 2020 (T3, approximately 3 years later), we successfully contacted 340 of them, representing 44.9% retention. The main drop-out reason was that some participants refused to join because of the surge in coronavirus cases in Hong Kong that was due to the pandemic at the time of the follow-up.

### 2.3. Variables and Measures

#### 2.3.1. IPV

The Chinese version of the abuse assessment screen (AAS) [20] was used to assess IPV experiences. The AAS has four items, namely participants’ experiences of physical violence, psychological violence, sexual violence, and fear because of IPV. Each item was rated either yes or no. Participants were asked to retrospectively report exposure to IPV before and during pregnancy at T1, during pregnancy and up to 4 weeks after childbirth at T2, and three years after childbirth at T3. Three items (physical violence, psychological violence, and sexual violence) were used in the current analysis, which is in line with previous work [14]. Participants who reported any positive response to any item were classified as having IPV exposure.

#### 2.3.2. Mental Health Conditions (Depressive Symptoms, Anxiety, and Stress)

Depressive symptoms were evaluated using the 10-item Chinese Edinburgh postnatal depression scale [21] (CEPDS; *α* = 0.84 at T1, *α* = 0.83 at T2, and *α* = 0.81 at T3). Anxiety and stress levels were measured using the 7-item anxiety subscale (*α* = 0.75 at T1, *α* = 0.59 at T2, and *α* = 0.69 at T3) and the 7-item stress subscale (*α* = 0.83 at T1, *α* = 0.83 at T2, and *α* = 0.80 at T3), respectively, of the Chinese version of the depression anxiety stress scale [22] (DASS). All items were rated from 0 to 3, with a higher score indicating higher levels of depression, anxiety, or stress.

#### 2.3.3. Health-Related Quality of Life (HRQoL)

Participants’ health-related quality of life (HRQoL) throughout the three periods was assessed using the 12-item Short-Form Health Survey [23] (SF-12; *α* = 0.87 at T1, *α* = 0.85 at T2, and *α* = 0.79 at T3). A total score ranges from 0 to 100, with a higher score representing better mental or physical HRQoL.

#### 2.3.4. Partner Involvement

Participants reported their partner involvement at both 4 weeks and 3 years after childbirth. Seven items were used, including partner involvement in assisting with daily housework and giving emotional support to the pregnant partner. These items were rated on a 4-point Likert scale. All items were summed, and a higher score indicated more partner involvement. Previous work also used these items to measure partner involvement [14]. The *α* was 0.87 at 4 weeks after childbirth and 0.90 at 3 years after delivery.

#### 2.3.5. Family Support

Participants’ perceived family support at both 4 weeks and 3 years after childbirth was assessed by the Chinese version of the multidimensional scale of perceived social support (MSPSS) [24], a 7-point Likert scale from 1 to 7. Four items were used to evaluate family support. We summed all four items, and a higher score indicated a higher level of family support. The *α* was 0.92 at 4 weeks after childbirth and 0.92 at 3 years after childbirth.

#### 2.3.6. Adverse Childhood Experiences

Exposure to childhood adversities was measured using an adaptation of the Adverse Childhood Experiences-International Questionnaire (ACE-IQ) [25]. The initial validation of the ACE-IQ in the Hong Kong context has been reported elsewhere [26]. Fourteen items were used to assess different domains of ACEs, such as physical abuse, sexual abuse, emotional abuse, physical neglect, emotional neglect, and domestic violence. Each item was dichotomized into “1” = “exposed” and “0” = “not exposed”. Items were summed to obtain an ACE score, and a higher score indicated a higher exposure to childhood adversities. The *α* was 0.61 in this study.

#### 2.3.7. Demographic Characteristics

Demographic information was collected, including age, education attainment, marital status, employment status, whether they were receiving any social security assistance from the government, monthly household income, chronic health conditions, and risk behaviors (gambling, smoking, or drinking).

### 2.4. Statistical Analysis

Percentages of IPV experiences from pre-pregnancy to 4 weeks after childbirth and 3 years after childbirth were calculated. As shown in Table 1, a violent relationship (VR) was defined as one with exposure to any IPV throughout the period from pre-pregnancy to 4 weeks after childbirth and 3 years after childbirth. A decreased violent relationship (DVR) was defined as one with IPV reported from pre-pregnancy to 4 weeks after childbirth but terminated at 3 years after childbirth. A nonviolent relationship (NVR) was defined as one with no IPV experiences throughout all the periods. The Chi-square test or F-test was used to compare distributions or scores of interested variables across the three categories.

A three-phase ordinal logistic regression analysis was used to explore the correlates of IPV changing categories. The dependent variable (i.e., the three IPV categories) was assigned in the order of “NVR < DVR < VR”, as the severity of violence increased in this order [14,27]. In phase 1, we conducted a forward stepwise ordinal logistic regression on all demographic characteristics. In phase 2, we entered the following variables individually after controlling for demographics: partner involvement at T2 and T3, and family support at T2 and T3. Finally, we added the following variables individually in phase 3 after adjusting for demographic characteristics: ACEs, risk behaviors, chronic illness, and health variables (mental health and quality of life). All analyses were conducted with SPSS 26.0.

## 3. Results

In total, 340 pregnant women participated in all three periods (retention rate = 340/758 = 44.9%). We compared those who took part in all assessments and those who did not. The results showed that those who completed all assessments had higher IPV percentages and higher exposure to ACEs than those who dropped out. In addition, those who dropped out had a higher level of education.

Of the 340 participants (M_age_ = 31.30, SD = 4.26), 22.9% reported IPV before pregnancy, 13.5% reported IPV during pregnancy, 14.7% reported IPV 4 weeks after childbirth, and 11.8% reported IPV 3 years after childbirth. As for IPV categories from pre-pregnancy until 4 weeks after childbirth and 3 years after childbirth (see Table 1), 11.8% of women experienced persistent IPV across time (the VR group), 20.6% of women reported decreased IPV (the DVR group), and 67.6% of women did not have any IPV experiences over these periods (the NVR group).

Table 2 shows differences between IPV groups in the study variables, including demographic characteristics, risk behaviors, chronic illness, ACEs, partner involvement, family support, and health outcomes. There were no statistically significant differences in demographic characteristics, risk behaviors, and chronic illness across the different groups except for whether they received social security assistance. Specifically, women in the VR group were more likely to receive social security assistance (*p* < 0.01). Furthermore, there were significant between-group differences in ACEs, mental health conditions, and quality of life (particularly the mental health component) (all *p* < 0.05). Women in the VR group reported more ACEs and poorer health consequences. In addition, they reported lower family support and lower partner involvement than those in the NVR group.

Table 3 presents the three-phase ordinal logistic regressions examining the associations between the study variables and IPV. In phase 1, receiving social security assistance was associated with an increased chance of experiencing more severe IPV (aOR = 6.44, 95% CI = 1.95, 21.18). In phase 2, higher partner involvement and higher family support were associated with a lower chance of experiencing severe IPV. In phase 3, after controlling for demographics, pregnant women reporting ACEs was associated with a higher possibility of them having severe IPV experiences (aOR = 1.25, 95% CI = 1.12, 1.39). Furthermore, depression at all three periods (aOR = 1.06 at T1, 1.08 at T2, and 1.07 at T3), stress at all three periods (aOR = 1.12 at T1, 1.17 at T2, and 1.09 at T3), and anxiety at 4 weeks after childbirth (aOR = 1.29) were associated with increased likelihood of the dependent variable, indicating that poorer mental health was related to an increased chance of experiencing more severe IPV. Better HRQoL, particularly in the mental dimension, was associated with lower odds of suffering more severe IPV (aOR = 0.96).

## 4. Discussion

The overall IPV prevalence decreased from pre-pregnancy to 3 years after childbirth. Yet, when we explored IPV categories across different periods, about 12% of women reported persistent IPV from pregnancy to 3 years after childbirth. These women were more likely to have had more ACEs and to have poorer health than women who did not experience IPV at all periods. Higher partner involvement and family support could decrease the chance of IPV.

Our results showed reductions in IPV prevalence from pre-pregnancy (22.9%) to 3 years after childbirth (11.8%); this should be cautiously interpreted because of our relatively low retention rate. The literature has inconsistent findings on IPV prevalence from pregnancy to postpartum. A follow-up study in Mexico City found that IPV slightly increased from pregnancy to postpartum [28]. However, another study showed decreased IPV from 32.2% during pregnancy to 25.3% after childbirth [29]. Different cultural contexts may explain these mixed findings. It is possible that women’s status differences and the degree of gender inequality within a specific culture differently impact the rates of IPV against pregnant women [30]. Future studies should explore IPV in other contexts. In addition, different methodologies (e.g., the mode and timing of IPV measurements) may result in a wide variation in IPV prevalence [13].

As for individual changes in IPV, a substantial number of women (11.8%) continuously experienced IPV from pregnancy to 3 years after childbirth. Our successful identification of different groups resonates with the literature suggesting that exploring individual variabilities in IPV is crucial [31]. Indeed, more recent work has supported that different women may experience different types of IPV [13,14]. Our evidence of continued IPV from pregnancy to 3 years after childbirth can alert healthcare providers that violence is likely to continue for years. Therefore, screening for violence during pregnancy and immediately after childbirth can help prevent future IPV, but it is insufficient to limit screening to only these periods. Our current study suggests that long-lasting screening till years after childbirth is critically important to protect the woman and her baby. Furthermore, we found that women in the VR group were at higher risk of poorer mental health. This finding is in line with existing reviews showing strong associations between IPV and mental disorders in pregnant women [32,33].

In total, 20.6% of women (the DVR group) reported IPV in the earlier stages but did not report IPV at 3 years after childbirth. Higher family support and partner involvement could buffer against IPV, which echoes the buffering model of social support. The model points out that stressful events may negatively influence those with little or no support. In contrast, stronger support systems could protect individuals, helping them to recover from stressful events [34]. Specifically, individuals receiving higher social support may be more likely to gain purpose in life and build a sense of control. These individuals could create a more adaptive narrative of stressful events (e.g., IPV in the current study) and thus alleviate their negative impacts [35]. It is noteworthy that we did not identify a sufficient number of new cases because of our limited sample. Future studies with more available data are recommended to screen for new cases because they may negatively affect the mental outcomes and behavioral problems of mothers and their children [36].

ACEs are potential risk factors for long-term IPV. Our findings indicated that women who experienced IPV were more likely to have more ACEs than women in the NVR group. The result held even after controlling for demographic covariates. A previous review has evidenced the negative impacts of ACEs on IPV [19]. Notably, this risk might be cumulative: the more ACEs, the higher the possibility of experiencing IPV might be in the future [37]. Moreover, IPV against pregnant women can increase the risk of postnatal child abuse. A longitudinal population-based study in Hong Kong found that IPV during pregnancy was associated with greater odds of both lifetime (aOR = 1.74) and preceding-year child physical abuse (aOR = 1.78) [38]. Another recent follow-up study in Japan supports this finding [39]. Therefore, prevention and intervention programs for treating past traumatic events are urgently needed to break the cycle of violence.

## 5. Implications

Individual differences in IPV across years after childbirth have been identified. Women in the VR group had poorer health conditions than those in the NVR group. This echoes the notion that IPV prevention and intervention programs assuming all women are at similar risk of victimization may be insufficient to provide individualized help [40]. Given that there are different categories of IPV from pregnancy to the postpartum years, we suggest future studies explore specific interventions and services for each category. Continuous clinical screening for violence during prenatal and postnatal care is needed. More attention should be paid to those who report IPV during pregnancy, as it is at high risk of recurring years after childbirth. When IPV is detected, timely and effective programs should be applied. A review of the effectiveness of interventions for IPV around the time of pregnancy found a significant decrease in IPV when applying home visitation programs and some multifaceted counseling interventions [41].

More family support and partner involvement could protect women from IPV experiences. Family-based counseling has great strengths in reducing various types of violence against women by increasing couples’ awareness and improving their relationships during pregnancy. A family-based counseling intervention in Iran based on the principles of GATHER (greet, ask, tell, help, explain, and refer) is a successful example that illustrates the promising effects of family-based counseling [42]. Additionally, the *For Baby’s Sake* whole-family approach is another promising program to work with parents from pregnancy to two years postpartum to break cycles of domestic violence and improve children’s outcomes. Specifically, this approach supports the entire family to end domestic violence, addresses the cycles of domestic violence and abuse (e.g., the impact of parents’ own childhood experiences of adversity), and seeks to improve parents’ mental health, infants’ emotional and social development, and parent-child attachment outcomes [43].

Experienced childhood adversity could distinguish IPV groups from those in the NVR group. This indicates that treating past traumatic events is critical to preventing future victimization. Trauma-informed care (TIC) is a comprehensive, multilevel approach that helps service providers and clients understand the impact of traumatic events on health indicators and behaviors [44]. For instance, perinatal care providers (e.g., perinatal nurses) are well-positioned to provide trauma-informed perinatal care, which could prevent or reduce the negative impact of ACEs (e.g., by decreasing the possibility of future IPV and breaking the cycle of violence) [45].

## 6. Strengths and Limitations

This study has several strengths, including a relatively long-term follow-up to identify different categories of IPV experiences throughout the years after childbirth and to test correlates of these categories. However, limitations should be acknowledged when interpreting our results. First, the use of self-report questionnaires may have led to reporting bias. Second, as a result of the low retention rate, the sample size of the study was relatively small, which prevented us from capturing a sufficient number of participants who only experienced IPV three years after childbirth but not before pregnancy, during pregnancy, or 4 weeks after childbirth. Future studies are suggested to explore the experience of this group of women, as partners may generally be less physically violent because it can harm the baby [46], while emotional IPV may not decrease [47]. Lastly, our results may not be generalizable to other contexts, as various cultural and ethnic factors may lead to different attitudes toward IPV. Future work may consider conducting a multicountry design to confirm and compare our current findings. Despite the limited generalizability of the results, our categories of IPV support findings on pregnant women from different countries [13] and other samples [40,48,49], suggesting that individual differences in IPV exist and even last for years after childbirth.

## 7. Conclusions

This 3-year study on pregnant women extends previous work by identifying different IPV changing patterns from pre-pregnancy until 4 weeks after childbirth and 3 years after childbirth. Three IPV categories were identified: VR, DVR, and NVR. Women in the VR group were more likely to have poor mental health problems and more ACEs. Higher partner involvement and family support were protective against IPV.

## Figures and Tables

**Table 1 ijerph-19-14397-t001:** Different Types of IPV Relationship According to IPV Exposure from Pre-Pregnancy to Three Years After Childbirth (n = 340).

IPV Categories	Period from Pre-Pregnancy to 4 Weeks after Childbirth	3 Years after Childbirth	n (%)
Violent relationship (VR) ^a^	IPV+	IPV+	40 (11.8%)
Decreased violent relationship (DVR)	IPV+	IPV−	70 (20.6%)
Nonviolent relationship (NVR)	IPV−	IPV−	230 (67.6%)

Note. IPV+ = reported IPV. IPV− = no IPV. VR was defined as having experienced any violence from pre-pregnancy until 4 weeks after childbirth and also at the follow-up (i.e., at 3 years after childbirth). DVR was defined as reporting IPV from pre-pregnancy until 4 weeks after childbirth, but not at the follow-up. NVR was defined as having no IPV experiences throughout all the periods. ^a^ We grouped those who did not report IPV experiences from pre-pregnancy to 4 weeks after childbirth but had IPV experiences at 3 years after childbirth into the VR category, as the number was small (0.3%).

**Table 2 ijerph-19-14397-t002:** Distributions and Mean Scores of the Study Variables, by Type of IPV.

	Overall (n = 340)	NVR (n = 230)	DVR (n = 70)	VR (n = 40)	*p*-Value by χ^2^ or F Test
Age at baseline (mean, SD)	31.30 (4.26)	31.36 (4.33)	31.01 (3.68)	31.45 (4.86)	0.815
Education attainment (n, %)					0.431
Lower secondary or below	44 (12.9)	30 (13.0)	11 (15.7)	3 (7.5)	
Upper secondary	104 (30.6)	67 (29.1)	20 (28.6)	17 (42.5)	
College/university or above	192 (56.5)	133 (57.8)	39 (55.7)	20 (50.0)	
Marital status (n, %)					0.137
Widow/separated/divorced	25 (7.4)	14 (6.1)	9 (12.9)	2 (5.0)	
Married	315 (92.6)	216 (93.9)	61 (87.1)	38 (95.0)	
Employment status (n, %)					0.737
Unemployment	118 (34.7)	83 (36.1)	22 (31.4)	13 (32.5)	
Employment	222 (65.3)	147 (63.9)	48 (68.6)	27 (67.5)	
Receiving social security assistance (n, %)					0.003 ^a^
Yes	12 (3.5)	3 (1.3)	5 (7.1)	4 (10.0)	
No	328 (96.5)	227 (98.7)	65 (92.9)	36 (90.0)	
Monthly household income (n, %)					0.229
Less than HKD15,000	32 (9.4)	19 (8.3)	11 (15.7)	2 (5.0)	
HKD15,000 to HKD39,999	132 (38.8)	87 (37.8)	26 (37.1)	19 (47.5)	
HKD40,000 or above	176 (51.8)	124 (53.9)	33 (47.1)	19 (47.5)	
Risk behaviors (n, %)					0.864
Yes (gambling, smoking, or drinking)	29 (8.5)	20 (8.7)	5 (7.1)	4 (10.0)	
No	311 (91.5)	210 (91.3)	65 (92.9)	36 (90.0)	
Chronic illness (n, %)					0.473
Yes	36 (10.6)	23 (10.0)	10 (14.3)	3 (7.5)	
No	304 (89.4)	207 (90.0)	60 (85.7)	37 (92.5)	
ACEs (mean, SD)	2.79 (2.08)	2.44 (1.88)	3.61 (2.34)	3.38 (2.23)	<0.001
Father’s involvement T2 (mean, SD)	21.87 (3.69)	22.30 (3.49)	20.64 (3.70)	21.53 (4.32)	0.003
Father’s involvement T3 (mean, SD)	24.13 (4.66)	24.00 (4.59)	24.47 (5.10)	24.28 (4.34)	0.749
Family support T2 (mean, SD)	23.65 (4.28)	24.13 (3.79)	22.80 (5.08)	22.38 (4.98)	0.010
Family support T3 (mean, SD)	23.34 (4.58)	23.70 (3.91)	23.40 (4.90)	21.18 (6.63)	0.005
Depression, CEPDS (mean, SD)					
T1	6.98 (4.50)	6.49 (4.49)	8.23 (4.58)	7.60 (4.01)	0.011
T2	4.31 (4.15)	3.83 (3.38)	5.51 (5.19)	4.93 (5.54)	0.007
T3	4.25 (4.46)	3.81 (3.92)	4.94 (4.99)	5.58 (5.93)	0.024
Anxiety, DASS (mean, SD)					
T1	2.77 (2.55)	2.54 (2.43)	3.51 (2.89)	2.80 (2.35)	0.020
T2	1.22 (1.59)	1.03 (1.25)	1.47 (1.80)	1.88 (2.51)	0.002
T3	0.76 (1.63)	0.60 (1.15)	1.20 (2.60)	0.95 (1.71)	0.020
Stress, DASS (mean, SD)					
T1	3.83 (3.35)	3.43 (3.23)	4.63 (3.36)	4.70 (3.63)	0.007
T2	2.58 (2.87)	2.18 (2.35)	3.26 (3.68)	3.68 (3.53)	0.001
T3	2.36 (2.99)	2.07 (2.53)	2.87 (3.58)	3.10 (4.01)	0.036
Health-related quality of life (mean, SD)					
PCS T1	46.29 (6.92)	46.56 (6.97)	45.80 (6.68)	45.62 (7.07)	0.586
MCS T1	48.34 (8.40)	49.33 (8.21)	46.21 (8.54)	46.39 (8.41)	0.007
PCS T2	50.93 (5.98)	51.48 (5.56)	49.97 (6.93)	49.46 (6.27)	0.047
MCS T2	52.51 (8.45)	53.40 (8.14)	51.08 (8.50)	49.90 (9.36)	0.015
PCS T3	53.14 (5.63)	53.18 (5.76)	52.80 (5.57)	53.48 (5.06)	0.817
MCS T3	52.83 (8.08)	53.14 (7.30)	53.39 (8.12)	50.08 (11.33)	0.070

Note. ^a^ using a Fisher’s exact test because of the small numbers of individual cells. *p*-Value less than 0.05 represents statistically significant differences. T1 = during pregnancy; T2 = 4 weeks after childbirth; T3 = 3 years after childbirth. SD = standard deviation. NVR = nonviolent relationship; DVR = decreased violent relationship; VR = violent relationship. PCS = physical composite score; MCS = mental composite score; CEPDS = Chinese Edinburgh postnatal depression scale; DASS = depression, anxiety, and stress scale; ACEs = adverse childhood experiences.

**Table 3 ijerph-19-14397-t003:** Adjusted Odds Ratios (ORs) and 95% Confidence Intervals (CIs) of Variables Obtained in the Multiphase Ordinal Logistic Regression using Type of Relationship as the Dependent Variable (n = 340).

Phase 1	Adjusted ORs		
Age at baseline	1.00 (0.94, 1.05)		
Education attainment			
Lower secondary or below	0.90 (0.42, 1.96)		
Upper secondary	1.31 (0.75, 2.29)		
College/university or above	1		
Marital status			
Widow/separated/divorced	1.03 (0.42, 2.54)		
Married	1		
Employment status			
Unemployment	0.63 (0.36, 1.10)		
Employment	1		
Receiving social security assistance			
Yes	6.44 (1.95, 21.18) **		
No	1		
Monthly household income			
Less than HKD15,000	1.15 (0.44, 3.01)		
HKD15,000 to HKD39,999	1.31 (0.76, 2.25)		
HKD40,000 or above	1		
Phase 2 (variables individually added, controlled for demographics)			
Overall father involvement T2		0.93 (0.87, 0.997) *	
Family support level T2		0.93 (0.88, 0.98) **	
Overall father involvement T3		1.02 (0.96, 1.07)	
Family support level T3		0.95 (0.90, 0.996) *	
Phase 3 (variables individually added, controlled for demographics)			
ACEs			1.25 (1.12, 1.39) ***
Risk behaviors			
Yes (gambling, smoking, or drinking)			0.98 (0.42, 2.25)
No			1
Chronic illness			
Yes			1.22 (0.59, 2.54)
No			1
Depression, CEPDS			
T1			1.06 (1.01, 1.12) *
T2			1.08 (1.03, 1.14) **
T3			1.07 (1.01, 1.12) *
Anxiety, DASS			
T1			1.08 (0.99, 1.18)
T2			1.29 (1.13, 1.48) ***
T3			1.11 (0.97, 1.27)
Stress, DASS			
T1			1.12 (1.05, 1.20) **
T2			1.17 (1.08, 1.26) ***
T3			1.09 (1.01, 1.17) *
Health-related quality of life			
PCS T1			0.98 (0.95, 1.02)
MCS T1			0.96 (0.93, 0.99) **
PCS T2			0.95 (0.92, 0.99) **
MCS T2			0.96 (0.93, 0.98) **
PCS T3			1.00 (0.96, 1.04)
MCS T3			0.99 (0.96, 1.01)

Note. T1 = during pregnancy; T2 = 4 weeks after childbirth; T3 = 3 years after childbirth. NVR = nonviolent relationship; DVR = decreased violent relationship; VR = violent relationship. The dependent variable was the type of relationship. It was ordinal: the threshold estimate “1” was the cut-off value between “nonviolent relationship” (NVR) and “decreased violent relationship” (DVR). The threshold estimate “2” was the cut-off value between “decreased violent relationship” (DVR) and “violent relationship” (VR). PCS = physical composite score; MCS = mental composite score; CEPDS = Chinese Edinburgh postnatal depression scale; DASS = depression, anxiety, and stress scale; ACEs = adverse childhood experiences. * *p* < 0.05. ** *p* < 0.01. *** *p* < 0.001.

## Data Availability

Data are available on request from the authors.

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
