# Peer review of "Changing Patterns of Intimate Partner Violence against Pregnant Women: A Three-Year Longitudinal Study"

_ijerph, 2022, doi:10.3390/ijerph192114397_

Round 1
Reviewer 1 Report
ABSTRACT
The research topic is of scientific and social interest.
In general, the article is correct, clarifies the purpose of the study and I consider that the topic is in line with the journal’s research objectives,
INTRODUCTION:
The study objective is well defined and identified in both the abstract and the introduction, well referenced. Clear the aims of the article
It’s been a good thing the references articles, the structure is well done, clear and focused.
MATERIALS, METHODS and RESULTS:
What is the meaning of XX hospital? And XXX University?
For the rest of this part:
The statistical treatment is correct. The number of patients in the study is correct. Well structured. Well selected the variables. Good structure.
Appropriate the tables, these are a proper clarification after the explication
DISCUSSION:
The conclusions and the discussion are well drawn and interesting. Appropriate extra notes in these parts.
An interesting line of research is observed
IMPLICATIONS, STRENGTHS AND LIMITATIONS:
Clarify points about the article, interesting point of view.
CONCLUSION:
An interesting line of research is observed, long term study well structured.
Reviewer 2 Report
I would like to thank you for giving me the opportunity to review this manuscript.
I would like to comment below on a number of improvements
The objectives of the study are 2 with a hypothesis for each one. It should be reviewed and, if possible, an objective should be made that covers the 2 objectives with the 2 hypotheses. Similarly, in the abstract, there is not record of these 2 objectives or at least of their main questions, only one of them is mentioned. This needs to be reviewed.
There are bibliographic references of more than 10, 15 and 20 years old. These should be revised and changed to more current ones.
Reviewer 3 Report
This is a valuable and well-presented study that makes good use of the data collected while acknowledging study limitations. Relevant literature is used appropriately with divergences in the evidence on this topic appropriately identified. The paper should be reviewed by the journal’s statistical reviewer but I concluded that the paper is of high quality and can be published with a couple of minor revisions:
It would be helpful to include a bit more information explaining how the confidentiality and safety of women completing the survey was protected at the different time-points.
The implications section refers briefly to an intervention for couples developed in Iran. It would also be worth referring to the ‘For Baby’s Sake’ programme delivered from pregnancy to families experiencing domestic violence and abuse in the UK. An evaluation of the pilot has been completed and the report is available:
Trevillion, K., Domoney, J., Ocloo, J., Heslin, M.Ling, X-X., Stanley, N., MacMillan, H., Ramchandani, P., Bick, D., Byford, S., Howard, LM. (2020) For Baby’s Sake: Final Evaluation Report. The Stefanou Foundation and King’s College London. www.forbabyssake.org.uk
